# Spinal Excitatory Dynorphinergic Interneurons Contribute to Burn Injury-Induced Nociception Mediated by Phosphorylated Histone 3 at Serine 10 in Rodents

**DOI:** 10.3390/ijms22052297

**Published:** 2021-02-25

**Authors:** Angelika Varga, Zoltán Mészár, Miklós Sivadó, Tímea Bácskai, Bence Végh, Éva Kókai, István Nagy, Péter Szücs

**Affiliations:** 1MTA-DE-NAP B-Pain Control Research Group, University of Debrecen, 4032 Debrecen, Hungary; miklos.sivado@anat.med.unideb.hu (M.S.); kokai.eva@med.unideb.hu (É.K.); szucs.peter@med.unideb.hu (P.S.); 2Department of Anatomy, Histology and Embryology, Faculty of Medicine, University of Debrecen, 4032 Debrecen, Hungary; meszarz@anat.med.unideb.hu (Z.M.); vegbence1997@gmail.com (B.V.); 3Division of Dental Anatomy, Department of Basic Medical Sciences, Faculty of Dentistry, University of Debrecen, 4032 Debrecen, Hungary; bacskai.timea@anat.med.unideb.hu; 4Department of Surgery and Cancer, Imperial College London, London SW7 ZAZ, UK; i.nagy@imperial.ac.uk; 5Department of Physiology, Faculty of Medicine, University of Debrecen, 4032 Debrecen, Hungary

**Keywords:** histone, epigenetic modification, superficial dorsal horn neuron, burn injury, neuropeptides, nociception

## Abstract

The phosphorylation of serine 10 in histone 3 (p-S10H3) has recently been demonstrated to participate in spinal nociceptive processing. However, superficial dorsal horn (SDH) neurons involved in p-S10H3-mediated nociception have not been fully characterized. In the present work, we combined immunohistochemistry, in situ hybridization with the retrograde labeling of projection neurons to reveal the subset of dorsal horn neurons presenting an elevated level of p-S10H3 in response to noxious heat (60 °C), causing burn injury. Projection neurons only represented a small percentage (5%) of p-S10H3-positive cells, while the greater part of them belonged to excitatory SDH interneurons. The combined immunolabeling of p-S10H3 with markers of already established interneuronal classes of the SDH revealed that the largest subset of neurons with burn injury-induced p-S10H3 expression was dynorphin immunopositive in mice. Furthermore, the majority of p-S10H3-expressing dynorphinergic neurons proved to be excitatory, as they lacked Pax-2 and showed Lmx1b-immunopositivity. Thus, we showed that neurochemically heterogeneous SDH neurons exhibit the upregulation of p-S10H3 shortly after noxious heat-induced burn injury and consequential tissue damage, and that a dedicated subset of excitatory dynorphinergic neurons is likely a key player in the development of central sensitization via the p-S10H3 mediated pathway.

## 1. Introduction

There is growing interest in epigenetic mechanisms of gene regulation as processes that regulate the development and maintenance of pain of various origins [1,2]. A wide array of epigenetic processes are being considered, including the modification of histones by acetylation [1,2] and phosphorylation [3,4]. Changes in gene expression are particularly important in spinal dorsal horn (SDH) neurons as, together with glial cells, they form circuitries that are pivotal for the first stage processing of nociceptive information [5,6]. Despite accumulating data from novel experimental strategies [7,8,9,10], our understanding of those circuitries, and hence of spinal nociceptive information processing, is still very limited. Therefore, the further thorough characterization of SDH neurons using new strategies is needed. The recently identified marker, phosphorylated (p) serine 10 (S10) in histone 3 (H3), an epigenetic tag that exhibits upregulation in a group of SDH neurons following various injuries and subsequent inflammation of peripheral tissues, may be used as part of such needed new strategies [3,4]. The phosphorylation of S10 in H3 is a permissive histone post-translational modification (PTM) that allows for the transcription of various early genes including *c-Fos*, which then act as transcription factors regulating the transcription of pain-related secondary response genes in the spinal cord of rodents [3,4]; however, c-Fos is expressed in a much broader set of SDH neurons [11,12]. Additionally, blocking the writer of phosphorylation of S10 in H3 specifically results in the lack of development of an important modality, heat hyperalgesia, and in prolonged pain associated with inflammation of tissues [3,4]. Hence, characterizing p-S10H3-expressing SDH neurons and the genes that are directly or indirectly controlled by that epigenetic tag would lead us to better understanding how SDH circuitries are involved in the development of hypersensitivity to heat in inflammatory conditions work.

Based on neurokinin-1 receptor (NK1R) expression, SDH neurons exhibiting upregulation in p-S10H3 expression following tissue injury/inflammation are regarded as putative projection neurons (PNs) with axons terminating in supraspinal areas [3]. While projection neurons of the SDH are all glutamatergic, SDH interneurons are either excitatory or inhibitory in function [5,13,14,15,16]. Based on their neurochemical properties, both excitatory and inhibitory neurons can further be divided into smaller, though somewhat overlapping groups of neurons; while excitatory cells are divided into somatostatin (SOM), neurotensin, neurokinin B, cholecystokinin (CCK), substance P (SP), gastrin-releasing peptide (GRP) and neuropeptide FF (NPFF)-expressing groups; inhibitory cells contain parvalbumin (PV), calretinin (CR), neuropeptide Y (NPY), neuronal nitrogen monoxide synthase (nNOS) or dynorphin (Dyn)/galanin [5,16,17,18,19,20,21,22,23,24,25,26]. Thus, in order to link the diverse SDH neuronal populations to the p-S10H3-mediated re-organization of spinal nociceptive circuitries, we have used burn injury to induce tissue damage and assessed the neurochemical properties and distribution of SDH neurons that exhibited consequential upregulation in p-S10H3 expression.

## 2. Results

### 2.1. p-S10H3 Expression is Upregulated in a Subpopulation of Ipsilateral SDH Neurons with Differing Rostrocaudal Density Following Burn Injury

While innocuous stimulation (37 °C for 2 min) did not induce the phosphorylation of S10H3 in the SDH (Figure 1a), in agreement with Torres-Perez et al. [4] there were numerous p-S10H3-labeled nuclei in the entire mediolateral extent of the ipsilateral laminae I–IIo 5 min after tissue damage induced by heat insult to the hind paw (60 °C for 2 min; Figure 1b; Table 1). All lumbar segments (L1–L6) contained p-S10H3-immunoreactive nuclei following burn injury, however, there was a significant difference in the number of neurons expressing p-S10H3 between rostral (L1–L3) and caudal (L4–L6) segments (Figure 1c). The highest number of p-S10H3 nuclei was encountered in the L6 spinal segment, which corresponded to the somatotopic origin of the stimulus applied to the distal part of the hind limb, and the number of p-S10H3 nuclei declined gradually towards the L1 segment (values correspond to the medians on the box plot; Figure 1c, Table 1; four to nine sections per segment from three animals). The estimated total number of neurons showing p-S10H3 in the lumbar region (L1 to L6) was 1452.2 ± 222 (four to nine sections per segment from three animals).

### 2.2. SDH Neurons Expressing p-S10H3 Following Burn Injury are in Close Apposition Mainly to CGRP-Containing Peptidergic Afferents

It is well-known that there is only limited overlap between isolectin B4 (IB4) and CGRP-positive primary afferent terminals in the superficial laminae due to their different laminar arborization derived from their differing sensory functions [28,29]. As expected, somata containing p-S10H3 restricted to lamina I and IIo were largely overlapping with the CGRP-positive area both in mice (Figure 2(a1)) and rats (Figure 2c), while p-S10H3-containing neurons in the superficial dorsal horn typically did not fall within the area occupied by IB4 terminals or within deeper laminae in either species (Figure 2b–d). Putative synaptic contacts from CGRP-immunopositive (IP) terminals could also be observed on the soma of p-S10H3-containing neurons (Figure 2(a2)). Of the total p-S10H3-positive nuclei, 84% (197/235) and 11% (27/235) were located within the CGRP- and IB4-immunoreactive (IR) area in mice, respectively (Figure 2d). Similarly, in rats, 84% of total p-S10H3 nuclei occupied the area of CGRP terminals (217/257), whereas only 11% were present in lamina IIi which is innervated by IB4 (29/257; Figure 2d).

In accordance with Saeed and Ribeiro-da-Silva et al. [30], we observed that the density of the CGRP fibers was higher in the medial part of the superficial layers, than in the lateral regions of the spinal cord in a transverse section (Figure 2a1, and c). Interestingly, these authors reported that CGRP immunoreactivity was also detected in the inner region of lamina II in the rat [30]. Therefore, colocalization studies in this study were performed in a 100 µm thick band measured from the surface of the superficial dorsal horn, where the majority of p-S10H3-expressing cells are surrounded by peptidergic CGRP fibers ( Appendix A). However, as peptidergic CGRP fibers rarely extended as deep as 100 µm into the grey matter of spinal cord, several p-S10H3-expressing cells fell outside the CGRP arborization territory (Appendix A). These outlier neurons, however, were clearly less abundant than their counterparts in the uppermost superficial layers (lamina I and IIo; Appendix A). As in mice, p-S10H3-positive nuclei fall mostly within the CGRP-positive band of laminae I–IIo, while IB4 immunoreactivity fibers did not arborize p-S10H3-containing neurons in the rat (Figure 2c,d).

### 2.3. S10H3 Phosphorylation Occurs Predominantly in SDH Neurons in Mice

Quantitative analysis revealed that 83.5% ± 8.9 of p-S10H3-expressing cells were neurons based on their Fox-3 (NeuN)-positivity (total number of double-labeled cells: 523; total number of p-S10H3+ cells: 626), whereas p-S10H3-IR cells accounted for 17.1% ± 3.7 of the neurons labeled with NeuN in lamina I and IIo (total number of double-labeled cells: 523; total number of NeuN+ neurons: 3048; number of sections: 20 from six wild-type CD1 mice; Figure 3).

The double immunolabeling of p-S10H3 with ionized calcium binding adaptor molecule 1 (Iba1) on another set of transverse spinal cord sections from wild-type mice revealed that Iba1-immunoreactive (IR) microglial cells that contained p-S10H3 in their nuclei were occasionally also visible in the superficial dorsal horn of the spinal cord (3.5% ± 0.8; total number of double-labeled cells: eight; total number of p-S10H3+ cells: 224; number of section: nine from three mice; Figure 3), suggesting that the involvement of spinal microglial cells in the somatosensory processing of noxious heat-induced burn injury and consequential tissue damage is negligible, at least in the acute phase of burn injury.

### 2.4. Only a Small Proportion of Projection Neurons Expresses Nuclear p-S10H3 Following Burn Injury

Noxious information carried by primary afferents is modulated by local neuronal circuits in the SDH before being transmitted by PNs to higher brain areas [14,15,31]. Thus, we hypothesized that PNs, which receive monosynaptic primary afferent input from both nociceptive C and Aδ fibers [14,15,31,32], might be the main target cells for the earliest response to burn injury in which p-S10H3 is upregulated in burn injury.

Projection neurons were scattered along the whole rostrocaudal extent of the lumbar spinal cord after retrograde labeling with cholera toxin b-subunit (CTb) from the lateral parabrachial nucleus (LPb). The side of CTb injection (right LPb) was contralateral to the side of burn injury (left hind leg). CTb-labeled neurons (i.e., PNs) were present in lamina I (Figure 4a), and also in the deeper layers (laminae III–IV; not shown) and were distributed bilaterally as expected [30], with the majority being contralateral to the side of the CTb injection (and ipsilateral to the noxious injury; Figure 4b). The average number of retrogradely labeled PNs was 39.7 ± 8.3 and 26.2 ± 7.7 on the ipsi- and contralateral sides (relative to the side of the burn injury), respectively (Figure 4b, n = 17 sections altogether from four rats). Three quarters of the CTb-labeled neurons on both sides (74.8% and 75.2% ipsi- and contralaterally, respectively) showed immunoreactivity for the antibody against the NK1 receptor (NK1R; 29.7 ± 8.2 ipsilaterally; 19.7 ± 5.9 contralaterally) further confirming their identity as PNs. Burn injury-induced p-S10H3 nuclei were present in only 16.3% of laminae I and III–IV PNs (6.5 ± 1.4 PNs with p-S10H3 nuclei; n = 17 sections from four animals; Figure 4b,c) and virtually all of these neurons also expressed NK1R (15.7% of the total laminae I and III–IV PNs). Only approximately 5% of the total SDH neuronal population expressing nuclear p-S10H3 following burn injury proved to be PNs (Figure 4c), while of the PN population, 16% contained p-S10H3 nuclei (Figure 4c).

### 2.5. S10H3 Phosphorylation Occurs Predominantly in Local Interneurons in Mice

Next, we wished to determine the SDH neuronal populations that show induced nuclear p-S10H3 upon burn injury. We selected characteristic markers that are abundant in excitatory and inhibitory SDH neuronal populations in laminae where the highest p-S10H3 expression was detected.

Pax-2 and Lmx1b are required for the gamma-amino butyric-acid (GABA)ergic and glutamatergic fates of spinal interneurons, respectively [6,8,33,34,35,36,37]. The double immunolabeling of p-S10H3 and Pax-2 in wild-type mice revealed that 24.0% ± 5.0 of p-S10H3-IR neurons was Pax-2 positive (36/150; n = 6 sections; Table 2; Figure 5a–c), while the double immunostaining of p-S10H3 and Lmx1b showed that 73.7% ± 6.8 (219/297; n = 9; Table 2; Figure 5d–f) of p-S10H3-immunopositive neurons co-expressed Lmx1b suggesting that the larger proportion of p-S10H3-expressing neurons appears to be excitatory in function (Table 2; Figure 5).

To further confirm our results, we applied yet another strategy and used anti-red fluorescent protein (RFP) antibody against tdTomato on lumbar spinal cord sections from vesicular gamma-amino butyric-acid (GABA) transporter (VGAT)- and vesicular glutamate transporter 2 (Vglut2):tdTomato mice, to determine the proportion of inhibitory and excitatory subsets of p-S10H3-expressing SDH neurons. The Vglut2 was selected as an excitatory marker, since virtually all spinal excitatory neurons express Vglut2 [38]. Inhibitory neurons were identified based on the presence of the VGAT, which is crucial for the loading and storage of GABA, as well as glycine in synaptic vesicles, and localizes exclusively in inhibitory axon terminals [39]. In this set of experiments, we found that only 15.9% ± 4.6 (67 out of 420; number of sections: 16 from two mice) of the total number of burn injury-induced p-S10H3 nuclei were confined within VGAT-positive neurons (Table 2; Figure 5g–i), while near half (44.5% ± 8.5; 128 out of 287; number of section: 16 from three animals) of them were located in glutamatergic neuronal somata as assessed by the expression of tdTomato in Vglut2:tdTomato mice (Table 2; Figure 5j–l). Note that the Vglut2 + VGAT population together does cover around two thirds of the neurons with burn injury-induced nuclear p-S10H3 expression. 

From these results, it appears that p-S10H3 upregulation in burn injury occurs in various types of interneurons, the majority of which are indeed excitatory.

### 2.6. Burn Injury-Induced S10H3 Phosphorylation Occurs Mainly in a Dynorphinergic Population of SDH Neurons in Mice

Since the majority of SDH neurons expressing p-S10H3 proved to be glutamatergic and near one fifth were GABAergic, we carried out double immunostaining for different neurochemical markers of both inhibitory and excitatory neuronal populations to determine their proportion in the subset of SDH cells responding to burn injury with nuclear p-S10H3 expression.

Among the well-defined excitatory interneurons, only somatostatin (SOM), substance P (SP) and GRP-positive cells are located in the superficial laminae [18,21,22,40,41] where p-S10H3 neurons are concentrated upon burn injury. Therefore, we examined the colocalization of these markers with p-S10H3 after burn injury. According to the literature [5,8,18,21], SOM-expressing cells account for ~60% of excitatory interneurons in the superficial dorsal horn and exhibit rather heterogenous populations expressing additional types of neuropeptides. In our experiments, the double labeling of p-S10H3 and SOM revealed that 60.2% ± 15 (47 out of 78; Figure 6a1–a3; Table 3) of p-S10H3-IR cells was SOM+, while 32.1% ± 5.9 of SOM-IR neurons expressed p-S10H3 (47 out of 146; data not shown).

Consistent with the findings that only 20% of the SP content of the dorsal horn is produced by local excitatory interneurons, SP-IR perikarya were fairly seldom encountered in laminae I–II, with abundant axonal arborization (Figure 6b1–b3). The most intense staining for *Tac1* mRNA was detected in neurons located outside the most superficial laminae, where very few p-S10H3-containing nuclei were observed and therefore there was virtually no overlap between *Tac1* mRNA and p-S10H3 protein (Figure 6(c1–c5)). GRP-IR neurons could not be identified with certainty due to the fact that GRP protein level in their cell bodies was below the threshold for detection with immunostaining (data not shown) as also confirmed by Barry et al. [41]. We have only detected very weak perikaryal staining in our immunofluorescent staining for substance P and GRP, despite trying several antibodies from different companies, and if anything, those neurons have been covered by the large amounts of primary afferents terminals in the most superficial laminae. Therefore, substance P and GRP-producing neuronal cell bodies were identified with riboprobes directed against *Tac1* (encodes SP) and GRP mRNAs in in situ hybridization (ISH) experiments combined with immunostaining for p-S10H3 (Figure 6c1–c5 and d). The most intense staining for *Tac1* mRNA was detected in neurons located in laminae II–III and also scattered evenly in the deeper dorsal horn (Figure 6c1–c5), and therefore there was virtually no overlap with p-S10H3 (Figure 6c1–c5). SDH neurons containing GRP mRNA never expressed pS10H3 following burn injury (Figure 6d).

Next, we selected five markers previously associated with inhibitory SDH neurons [42]: parvalbumin (PV), calretinin (CR), neuropeptide Y (NPY), neuronal nitrogen monoxide synthase (nNOS) and prodynorphin (Pdyn). Sections from wild-type mice were reacted with p-S10H3, one of the above listed inhibitory markers and the neuronal cell body-specific NeuN to reveal the distribution of the p-S10H3-expressing neurons among the inhibitory SDH neuronal populations. Due to species incompatibility in the cases of calretinin and parvalbumin, NeuroTrace fluorescent Nissl stain (Nissl) was applied instead of NeuN (Table 4). Representative images of sections derived from wild-type mice immunostained for these inhibitory markers are shown in Figure 7, while quantitative analysis of the proportions of markers in neurons with p-S10H3 nuclei is summarized in Table 3. Neurons in laminae I–IIo expressing p-S10H3 upon burn injury showed the highest co-expression (34.2% ± 3.3; 95/277; n = 7 sections from three wild-type mice) with prodynorphin and 62.9% ± 3.3 of the Pdyn-IR neurons expressed nuclear p-S10H3 (95/151; n = 7 sections from three wild-type mice; Figure 7e1–e3). To improve the detection of dynorphinergic somata, we repeated the quantification on sections from Pdyn:enhanced green fluorescent protein (EGFP) transgenic hybrid mice. We have previously validated this hybrid and showed that over 90% of EGFP+ cells in sections from Pdyn:EGFP mice were Pdyn-IR neurons (Appendix A). In these sections, 38.6% ± 4.3 (78/202; n = 10 sections from three animals) of p-S10H3-immunoreactive nuclei belonged to dynorphinergic neurons, which is consistent with the data obtained above in wild-type animals.

In accordance with Boyle et al. [42], the great majority of dynorphin-expressing cells in laminae I–IIo also expressed galanin (89.1%; 90 out of 101), whereas only 57.3% of the galanin-expressing cells co-labeled with dynorphin (90 out of 157; 7 sections from two mice; data not shown). The other notable subpopulation of SDH cells that expressed nuclear p-S10H3 was that of calretinin neurons (8.3% ± 2.3), whereas 12.6% ± 1.7 of calretinin-expressing cells showed p-S10H3-IR (Figure 7b1–b3 and Table 3). The proportion of p-S10H3+/CR+ double-labeled neurons that were inhibitory and excitatory were 36% ± 1.4 and 75% ± 2.6, respectively, based on their co-labeling with tdTomato-specific anti-RFP in VGAT- or Vglut2:tdTomato mice (Appendix A). The other three inhibitory markers (parvalbumin, NPY, nNOS) showed very limited overlap, if any, with p-S10H3 (Figure 7a1–a3, c1–c3, d1–d3; see in Table 3). However, the SDH region where p-S10H3 nuclei were concentrated showed a high density of NPY-positive processes and bouton-like profiles (Figure 7c1–c3).

Taken together, these results indicate that p-S10H3-expressing nuclei mostly belong to SOM+ and Pdyn+ cells. As stated above, SOM+ neurons exhibit rather diverse populations expressing various types of additional neuropeptides including Pdyn [5,8,18,21]. While our own observation that 23.3% of p-S10H3+/Pdyn+ double-labeled neurons (5 out of 21 in three sections) was SOM+ in Pdyn:EGFP mice (unpublished data) supports this, we focused our interest on the distinct population of Pdyn+ neurons that showed p-S10H3 upregulation following burn injury-induced tissue damage and did not try to separate the SOM/Pdyn double positive group.

### 2.7. The Highest Proportion of p-S10H3 Nuclei Appears in Excitatory Dynorphinergic Neurons Following Burn Injury

The above experiments confirmed that the majority of the p-S10H3-expressing neuronal population belonged to an excitatory pool of interneurons as p-S10H3 upregulation caused by noxious heat-induced burn injury and tissue damage occurred to a greater extent in Lmx1b-IR SDH neurons (73.7%; see Table 2). While dynorphinergic neurons are mainly GABAergic, excitatory interneurons have also been demonstrated to express Pdyn in the mouse [42,43,44], thus we wanted to check if Pdyn neurons expressing p-S10H3 after burn injury belong to the excitatory group. 

To address this issue, we determined the proportion of p-S10H3-expressing Pdyn neurons that were immunoreactive to Pax-2 and Lmx1b on sections from Pdyn:EGFP mice (n = 3 animals; Figure 8). As expected, we found that only 22.2% ± 3.3 of p-S10H3-expressing dynorphinergic neurons (10/45; Figure 8a) were Pax-2-positive, while the majority of them (83.3%; 30/36; Figure 8b) had Lmx1b-IR in laminae I-IIo following burn injury. The proportion of p-S10H3-expressing Pdyn+ cells with Lmx1b-IR was significantly higher than the proportion of p-S10H3-expressing Pdyn+ neurons with Pax-2-IR (*p* = 0.026; 3–6 sections from three mice).

Using an alternative strategy, of p-S10H3+/Pdyn+ double-labeled neurons, 20.4% ± 6.4 (10 out of 49; n = 6) were inhibitory and 47.2% ± 13.3 (34 out of 72; n = 7) were excitatory based on their co-labeling with tdTomato-specific anti-RFP signals in VGAT- or Vglut2:tdTomato mice (Appendix A). This result was consistent with our previous findings obtained with Pax-2 and Lmx1b antibodies, although the proportion of excitatory p-S10H3-expressing Pdyn-IR cells based on findings in the Vglut2:tdTomato mice appears to be underestimated as compared to the value calculated by Lmx1b positivity.

In summary, these results strongly suggest that the excitatory, rather than inhibitory subset of dynorphinergic neurons contributes to nociceptive processing mediated via p-S10H3 upregulation following noxious heat.

## 3. Discussion

Transcriptional and translational modifications in SDH neurons are pivotal in the development and maintenance of long-term pain conditions. Activity- and stimulus-dependent changes in gene expression regulated by epigenetic mechanisms (e.g., acetylation, phosphorylation, methylation, ubiquitylation, etc., of histone proteins) lead to alterations in neuronal activity [1,2,45,46]. The significance of histone 3 (H3) acetylation in the maintenance of pathological pain conditions has already been emphasized in previous studies [2,47]. Histone H3 phosphorylation at serine 10 in the spinal cord, however, has only gained attention recently as a potential player in nociceptive sensitization in several animal models of inflammatory pain [3,4], achieving this by a cell-type dependent permissive transcriptional program in post-mitotic neurons [3,46,48]. Superficial dorsal horn (SDH) neurons participating in pain processing, however, show considerable heterogeneity [16,49], thus our goal was to determine the segmental distribution of p-S10H3 in the lumbar segments of spinal cord in the mouse and also to identify subpopulations of SDH neurons that express p-S10H3 and also following noxious thermal stimulation.

Several technical considerations must be highlighted first. Noxious stimuli (60 °C; capsaicin or formalin), which can cause dramatic tissue damage, increased the number of the neurons expressing p-S10H3 [3,4] (see also in Figure 1). In contrast, brief noxious or innoxious stimuli (i.e., 37 °C; moderate pressure or noxious cold) did not result in any sign of central sensitization as assessed by p-S10H3. Thus, this PTM, at least in the spinal cord, seems to be linked to noxious stimulation. Burn injury-induced tissue damage was not assessed in this study, since the original article [50] proved that this model of scalding type thermal stimulus causes severe burn injury, damaging both the epidermis and the dermis of the hind paw. Consequently, the term “tissue damage” through the text refers to the massive histopathological alteration of the skin of the exposed hindlimb in response to the noxious thermal stimulus [50]. It must be noted that the anti-p-S10H3 antibody raised in rabbit produced a dotted staining pattern within the dorsal column of the white matter, however, this did not in any way interfere with the identification of p-S10H3-IR nuclei in the grey matter of the spinal cord. Labeling in the white matter is probably due to non-specific binding of the antibody that is unlikely to be burn injury-related phenomena as it could be observed in naive animals, too. While earlier reports indicated that p-S10H3 is exclusively expressed in neurons [3,4], we now found that NeuN-immunoreactive p-S10H3-expressing cells made up only 83.5% of the total p-S10H3 population in the SDH. Some p-S10H3-expressing cells exhibited strong Iba1-IR (3.5%) in our experiments, but in line with Torres-Perez et al. [4], we also confirmed that astrocytes do not show p-S10H3 upregulation after burn injury (data not shown). Since this dynamic epigenetic tag on histone H3 tail is rather transient in nature [3,4], to prevent the activity of the endogenous phosphatases and so that S10H3 would remain to be phosphorylated for a while, a phosphatase inhibitor was added to the paraformaldehyde (PFA) solution during perfusion. Therefore, we assume that our analysis provided unbiased estimates of the proportions of p-S10H3-positive nuclei among different pools of cells. Still, despite of our efforts, the identity of the remaining ~10% of p-S10H3-expressing cells remains to be elucidated.

It is well documented that c-Fos is expressed by NK1R-positive neurons following noxious heat [12,51,52] a high proportion of which are lamina I PNs relaying nociceptive information to supraspinal brain areas [12,53]. Thus, it is reasonable to assume that some of these heat-sensitive PNs might show p-S10H3-mediated transcriptional changes upon burn injury-associated tissue damage. While nearly all lamina I spinoparabrachial neurons express c-Fos upon noxious thermal stimuli [12,54,55], Tochiki et al. [3] found that only 17% of lamina I NK1R-positive neurons contained nuclear p-S10H3. Since NK1R is also expressed by lamina I interneurons [29,40,56] to assess PNs showing p-S10H3, we retrogradely labeled them from the LPb, where most (85%) PNs send collaterals in rodents [12,53]. While our finding that 16% of CTb-labeled projection neurons contained p-S10H3 fits well the results of Tochiki and his coworkers [3], considering also the 5% share of PNs from the total p-S10H3 SDH population, it is likely that only a small subset of heat-sensitive PNs are directly affected by transcriptional changes induced by burn injury. It cannot be excluded, however, that the proportion of p-S10H3-IR projection neurons which directly participate in processing thermal stimulus via S10H3 phosphorylation was underestimated, since only PNs projecting to the LPb had been taken into consideration in this study.

Specific ablation of excitatory interneurons in the SDH caused a complete loss of pain and itch sensation, indicating that different subsets of dorsal horn excitatory interneurons contribute to tissue- and nerve injury-induced hypersensitivity [57]. Consistent with our previous study [4], the present findings apparently provide further evidence that the vast majority of the p-S10H3-positive SDH neurons are excitatory based on their Vglut2 content along with a fair contribution of VGAT-positive inhibitory SDH neurons. These results were verified by immunolabeling for antibodies against to the cell-fate determining transcription factors as well (Pax-2 and Lmx1b), which label GABAergic and glutamatergic neurons with non-overlapping manner [6,8,33,34,35,36,37]. The major transcription factors of SDH neurons (Pax-2 and Lmx1b) cover the entire population of neurons showing p-S10H3 upregulation, while the Vglut2 + VGAT population cover around two thirds of the neurons with burn injury-induced nuclear p-S10H3 expression, leaving ~30% of cells to be unclassified. This discrepancy might be explained by the fact that the VGAT^cre^ and Vglut2^cre^ knock in mice used in this study were generated by inserting an internal ribosome entry site (IRES)-Cre cassette after the VGAT and Vglut2 stop codons, respectively [58]. The IRES-dependent downstream gene expression is about 50–80% lower than the expression of the upstream gene [59]. Thus, it is reasonable to speculate, that several VGAT+ or Vglut2+ neurons in these transgenic mice were less able to produce Cre recombinase due to the IRES strategy. Consequently, in hybrid mice (VGAT:tdTomato and Vglut2:tdTomato) the weak Cre expression was not sufficient for activating the reporter tdTomato expression in certain neurons that were unidentified. Thus, it is possible that these VGAT+ or Vglut2+ neurons were activated by the noxious stimulus and expressed p-S10H3, yet they remained tdTomato-negative (VGAT- or Vglut2-negative).

Neuropeptide signaling plays a significant role in spinal somatosensory processing [5,17,18,19,20,21,43,60]. Since neurokinin B-, CCK- and neurotensin-expressing cells are all found in the lamina IIi/III border and largely overlap with protein kinase C gamma (PKCγ) [5,19,21], their contributions to p-S10H3 upregulation after burn injury are questioned due to their different laminar distribution. Among the excitatory neurons, somatostatin (SOM), substance P (SP) and GRP-producing neurons are present in the superficial laminae [5,18,21,22,41], however, only SOM co-expressed considerably with p-S10H3 after noxious heat (60%). Taking into consideration that SOM-expressing cells account for ~60% of the excitatory interneurons in the SDH and exhibit a rather heterogenous population expressing multiple types of neuropeptides [5,8,18,21,22], it can be assumed that a more discrete subgroup of excitatory interneurons within this largely heterogenous SOM+ pool takes part in p-S10H3 upregulation. According to our preliminary/unpublished observation, nearly a quarter of the p-S10H3+/Pdyn+ double-labeled neurons were SOM+ as well. This may raise the possibility that, in contrast to a previous description [8], SOM+ neurons showing p-S10H3 upregulation might include the excitatory Pdyn+ population also, at least partially.

By combining ISH for *Tac1* or GRP RNAs with immunostaining for p-S10H3, we provided evidence that none of these excitatory neurons are involved in the p-S10H3-mediated pathway following burn injury and consequential tissue damage. In contrast to Gutierrez-Mecinas et al. [18,22], we found that the most intense staining for *Tac1* mRNA was detected in neurons distributed throughout the deeper dorsal horn (III–IV). It was confirmed by Xu and his coworkers [37] that most *Tac1* mRNA expressing cells occupy deep laminae, with the remaining few being scattered through superficial laminae in the spinal cord of young (P7) animals. Additionally, ISH data from GENSAT database (www.gensat.org/imagenavigator.jsp?imageID=2738; accessed on 5 October 2020) also enforces our observation. The discrepancy might be due to the use of transgenic animals [18,22] in which the *Tac1* gene might have been transiently turned on/off at any stage of development, resulting in ratios different from the ones in wild type animals.

The above-mentioned neuropeptides have earlier been identified as important players in the processing of different sensory modalities, including nociception. Somatostatin was implicated in the itch pathway via disinhibition [60] and also in acute mechanical pain [43], whereas GRP-mediated signaling is also associated with spinal itch transmission [22], although the latest results show that specific ablation of GRP neurons in the spinal cord has no influence on either itch or pain processing [41].

Other neuropeptides such as neuropeptide FF (NPFF), which has lately been linked to a neurochemical subgroup among the excitatory neurons, and account for 6% of excitatory interneurons and about 30% of those co-expressed with phospho (p)-extracellular signal-regulated kinase (ERK)1/2 in lamina I and II of the mouse spinal cord after noxious heat stimulus [17], was not examined in this study. However, we found that the great majority (84.7%) of the p-ERK1/2 immunopositive nuclei exhibited p-S10H3 immunolabeling, but only half of the p-S10H3-IR neurons co-expressed with p-ERK1/2 [4], indicating that these regulatory molecules could be phosphorylated and thus, activated in different subsets of dorsal horn neurons that do not overlap completely. As a consequence, the proportion of NPFF+ cells within the excitatory neuronal population that responded to noxious temperature would be around 2% (i.e., one third of the 6% [17]) suggesting that they are unlikely to contribute notably to the development of central sensitization via the p-S10H3 mediated pathway. Further findings that 85% of NPFF+ neurons were SOM positive and 40% contained GRP mRNA [17] also support our previous assumption.

Polgar et al. [23] demonstrated that parvalbumin+ neurons do not participate on c-Fos/p-ERK upregulation following noxious stimulation, whereas NPY+ and nNOS+ cells responded to a variety on noxious stimuli. Despite receiving direct primary afferent input from nociceptors, at least in the rat [23], nNOS- and NPY-containing inhibitory SDH neurons showed negligible nuclear p-S10H3 upon burn injury. While earlier studies agreed that calretinin-IR neurons are glutamatergic [16,20], recently, around one fourth of inhibitory dorsal horn neurons in the mouse have been reported to be calretinin-positive [8,42,61]. In the present study, we found that 8.3% of p-S10H3-expressing neurons showed calretinin-immunopositivity in response to burn injury and three quarters of these neurons were glutamatergic, as assessed by immunohistochemistry (IHC) in Vglut2:tdTomato mice. Dynorphinergic (Pdyn) neurons had the largest share of p-S10H3-expressing neurons following burn injury both in wild-type and Pdyn:EGFP transgenic mice (~63% and ~42%, respectively) suggesting their key involvement in the development of heat hyperalgesia after burn injury. While we did not make an attempt to confirm it, calretinin+ neurons expressing p-S10H3 are unlikely to be overlapping with the excitatory dynorphin+ population, as suggested by the recent findings of Gutierrez-Mecinas et al. [20].

Inhibitory neurons produce c-Fos or pERK1/2 more readily following noxious heat [11,18] than their excitatory counterparts. In contrast to this observation, on sections from VGAT- or Vglut2:tdTomato transgenic mice, we found that excitatory calretinin+ or Pdyn+ neurons showing p-S10H3 in their nuclei were more numerous in number than inhibitory ones. One possible explanation for the discrepancy between our findings and that of Gutierrez-Mecinas et al. [18] is that ~60% of p-S10H3-positive neurons co-expressed c-Fos, whereas ~20% of c-Fos-positive neurons co-expressed p-S10H3 as quantified by Torres-Perez et al. [4]. The remaining p-S10H3-positive neurons that lack c-Fos (~40%) presumably belongs to the excitatory subset of SDH neurons.

Early works demonstrated that complete Freund’s adjuvant (CFA) injection resulted in prodynorphin and NK1R upregulation through the activation of the mitogen-activated protein kinase (MAPK)/ERK pathway, indicating Pdyn neurons might contribute to acute inflammatory pain hypersensitivity [62,63]. Moreover, many of the dynorphin-containing neurons were shown to express c-Fos after noxious stimulation [64,65]. In accordance with several studies [42,43,44,66], we found that Pdyn neurons are abundant in laminae I–IIo at the L4–L5 level in mice. Although dynorphinergic neurons in these laminae were found to be mainly inhibitory based upon their Pax-2-IR [37,42] or their GABA-IR [44], an excitatory subset of Pdyn cells was also shown to be present in the SDH of spinal cord. In our experiments on L4–L5 sections from Pdyn:EGFP mice approximately 83.3% of Pdyn+ SDH neurons that contain p-S10H3 were excitatory based on their Lmx1b-IR. In line with observations in rat [44], we found that non-GABAergic Pdyn neurons (i.e., the Lmx1b+ population) are uniformly distributed throughout the entire superficial laminae (laminae I–IIo) in the mouse spinal cord. Pdyn neurons located more deeply in the dorsal horn, however, were all Pax-2-immunonegative in our experiments, which is in line with earlier findings [42,44]. Accordingly, a recent transcriptomics-based study also identified a cluster of excitatory SDH neurons (Glut14) containing Pdyn [8], although concentrated in deeper laminae. 

Galanin-containing neurons have been implicated in cold allodynia [67], thus in line with the recent reports of different inhibitory interneuron populations to be modality-specific (e.g., pain and itch) [60,68]; our results suggest that the dynorphinergic excitatory group might show a preference to respond to tissue damage induced by burn injury. Inhibitory Pdyn neurons probably respond to a wider range of stimuli, being also involved in spinal itch processing [60] or the spontaneous development of mechanical allodynia [43]. Some projection neurons in lamina I, including heat-selective ones [43], are considered to be dynorphinergic [69,70], thus the few PNs showing p-S10H3 upon burn injury might be Pdyn+. Further studies will be needed to address this. Noxious thermal or chemical stimulation provokes the activation of the endogenous pain control system, involving neuropeptides such as dynorphin and enkephalin via as-yet-unidentified molecular/epigenetic mechanisms [63,64,71]. Our findings that excitatory dynorphinergic cells are activated through the p-S10H3 mechanism suggest that this response to burn injury might be part of the activation of the endogenous pain control system. Whether this is a direct activation of SDH dynorphinergic neurons or happens through a spinal cord–brainstem–spinal cord loop remains to be elucidated.

## 4. Materials and Methods

### 4.1. Animals and Ethical Considerations

Experiments were performed on adult (ranging from 2–4 months old) rats and mice of both sexes (see details below in Section 4.2.). Experiments were approved by the Animal Care and Protection Committee at the University of Debrecen (No.: 23-1/2017/DEMÁB) and were performed in accordance with the European Community Council Directives and the IASP Guidelines. Animals were housed individually in a temperature-controlled colony room and maintained on a 12h/12h light/dark cycle. Food and water were provided ad libitum. An Ai14(RCL-tdT)-D reporter line expressing the red fluorescent protein tdTomato in a Cre-dependent manner was purchased from The Jackson Laboratory (JAX; Bar Harbor, ME USA; Stock No. #021875) [72]. The Cre-mediated excision of the STOP cassette was driven by the Cre recombinase expressed in VGAT^cre^ or Vglut2^cre^ mice in a particular subset of neurons (JAX; #016962, #016963) [58]. In the resulting hybrid mice (VGAT:tdTomato and Vglut2:tdTomato), all VGAT- or Vglut2-containing neurons showed strong somatic tdTomato expression. Pdyn-IRES-Cre mice (Pdyn^cre^; JAX #027958) which express Cre recombinase under the direction of the Pdyn (prodynorphin) promoter [73] were crossed with Rosa26-LSL-Cas9 (JAX #026175) knock in mice having Cre-dependent expression of CRISPR associated protein 9 (cas9) endonuclease and EGFP directed by a CAG promoter [74]. The resulting offspring (Pdyn:EGFP) should have EGFP in all neurons that have expressed Pdyn. Wild-type CD1 mice and Wistar rats were purchased from Charles River Laboratories (Wilmington, MA, USA).

### 4.2. Burn Injury Model

An animal model of burn injury that has been described by White et al. [50], and also by Torres-Perez et al. [4], was applied. Briefly, after inducing deep anesthesia with sodium pentobarbital (50 mg/kg intraperitoneal) the left hind leg was immersed into 60 °C water for 2 min. In control cases, the hind legs were immersed into 37 °C for 2 min. Five minutes after the burn injury, animals were transcardially perfused with a 4% paraformaldehyde (PFA) solution that contained 0.2% (wt/vol) sodium fluoride (NaF), a phosphatase inhibitor (Tochiki et al., 2016). A total of eight wild-type CD1 (3 males and 5 females), three Pdyn:EGFP (2 males and 1 female), two VGAT:tdTomato (2 males), three Vglut2:tdTomato hybrid (1 males and 2 female) mice and six Wistar rats (only males) were used altogether.

### 4.3. Retrograde Labeling of SDH Projection Neurons

For the retrograde labeling of SDH projection neurons, stereotaxic surgery was performed as detailed in previous studies [12,53]. Four Wistar rats (250–300 g) were deeply anaesthetized with a mixture of ketamine and xylazine (100 mg/kg and 10 mg/kg, respectively). 200 nanoliters of a cholera toxin b-subunit (CTb; List Biological Labs, Campbell, CA, USA) solution (1% wt/vol) was injected into the right LPb during 5 min with a 25 uL Hamilton pipette (Hamilton, NV, USA). The following coordinates were used to target the right LPb: 9.12 mm caudal to the Bregma, 2.4 mm right to the midline. The depth was 5.8 mm from the surface of the pia mater [75]. After a survival period of one week, animals were deeply anaesthetized with sodium pentobarbital (50 mg/kg intraperitoneal), exposed to burn injury and transcardially perfused. The side of CTb injection (right LPb) was contralateral to the side of burn injury (left hind leg). L4–L5 segments were used for immunofluorescent staining with the appropriate antibodies (CTb, p-S10H3 and NK1R). 

### 4.4. Immunoperoxidase Staining

For the evaluation of the rostrocaudal distribution of p-S10H3, the whole lumbar spinal cord (L1–L6) was removed and sectioned with a vibrotome (Leica VT 1000S; Leica, Wetzlar, Germany) at 80 µm thickness. Spinal cord segments were identified using vertebral landmarks [27]. After quenching endogenous peroxidase activity with 2% H_2_O_2_, an antibody raised against p-S10H3 in rabbit was added to the samples (1:2000; overnight; 4 °C). An ImmPress horseradish peroxidase (HRP) reagent kit (Vector Labs; Burlingame, CA, USA; 2 h; RT) and then a 3,3′-diaminobenzidine (DAB) peroxidase substrate kit (Vector Labs) were used for visualization of the p-S10H3 signal.

### 4.5. Double and Triple Immunofluorescent Staining

For the neurochemical characterization, L4–L5 segments were carefully dissected, post-fixed for 3 h in PFA and sectioned with a cryostat (Leica CM3050 S; Leica) at 100 µm thickness. Blocking was carried out with 5% normal donkey or goat serum (NDS/NGS; Sigma; St. Louis, MO, USA) for an hour followed by incubation in primary (2 days; 4 °C) and secondary (2hrs at room temperature, or overnight at 4 °C) antibody mixtures. All antibodies were diluted in Na-borohydride-containing phosphate buffer (0.1M PB) supplemented with a 0.3% Triton-X 100 and 1% (*v*/*v*) appropriate serum. Three 10-minute washes in 0.1 M PB were performed after incubation with all solutions except after the blocking step. Details of the primary antibodies applied in this study are presented in Table 4. Species-specific secondary antibodies were raised in donkey or goat and conjugated to Alexa Fluor-488, 555 and 647 (Invitrogen). At the end of the protocol, cell nuclei-specific 4′,6-diamidino-2-phenylindole (DAPI) was added to help determine the laminar boundaries based upon the orientation and density of the nuclei. Sections were mounted in a Hydromount medium (National Diagnostics; Brandon, FL, USA) and confocal images were scanned with Olympus FV3000 confocal systems (Tokyo, Japan).

### 4.6. Combination of In Situ Hybridization with Immunofluorescence (ISH/IF)

ISH was performed using a digoxigenin(DIG)-11-uridine diphosphate (UDP)-labeled cRNA antisense probe for substance P (bases 98–753 of tachykinin 1 (*Tac1*) mRNA, NCBI accession NM_009311.3) and for GRP (bases 219–643 of GRP mRNA, NCBI accession NM_175012). For the generation of the DIG-labeled RNA probes, we followed the guidelines from Roche (Basel, Switzerland; DIG application manual for non- radioactive in situ hybridization, 4th eds, Chapter 5.). Briefly, total RNA was isolated from the whole brainstem of a wild-type (C57Bl/6) mouse, and a cDNA template was generated using random primers following gene-specific PCR. Antisense RNA probes were then generated by in vitro transcription using T7 RNA polymerase on the PCR template. Two anesthetized wild-type CD1 mice were exposed to burn injury and treated with the same fixation procedure described above at the immunohistochemistry. The notable difference was that all surfaces and solutions contacted with the samples or the RNA probes were pretreated with diethyl-pyrocarbonate (DEPC, Sigma–Aldrich) or RNaseZap (Thermo Fisher Sci. Inc. (Waltham, MA, USA).

Free-floating 80-μm-thick transverse spinal cord sections were digested with Proteinase K (2 μg/mL), and then incubated with *Tac1*- or GRP-specific riboprobes (1:200) in hybridization solution for overnight at 60 °C.

After saline–sodium citrate buffer (SSC) stringency washes and RNase A treatment (16 μg/mL, for 5 min, at 37 °C), anti-digoxigenin antibody (produced in sheep, 1:1000, Sigma–Aldrich) was added to the slides together with the anti-p-S10H3 antibody raised in rabbit (1:500; overnight at 4 °C). After incubation with species-specific secondary antibodies (conjugated to Alexa Fluor-488 or 555; 1:500; overnight at 4 °C), cell nuclei-specific DAPI was added (5 µg/mL; 15 min) and sections were mounted in a Hydromount medium. Confocal images were scanned with Olympus FV3000 confocal systems.

### 4.7. Imaging and Quantification

DAB visualized on transverse sections were photographed with a digital camera (Olympus DP72) attached to an upright microscope (Olympus BX51). To evaluate data in an unbiased way, the analysis of the images was carried out by two independent researchers who were “blind” to the experiment. To evaluate the mean number of p-S10H3-positive neurons, the total number of DAB-stained p-S10H3+ nuclei in the SDH was counted in each 100-µm-thick transverse section and then averaged by spinal segments (L1–L6; Table 1).

To determine the colocalization, confocal image z-stacks consisting of up to 45 optical images at 0.5 um z-separation were scanned with ×40 (UPlanFLN, Olympus, N.A. 1.3) or ×60 (PlanApoN, Olympus, N.A. 1.4) lens from the entire width of the superficial dorsal horn with an Olympus FV3000 confocal system. In all cases, cell nuclei showing a DAPI signal were used as a reference for determining the full thickness of the slices.

To determine p-S10H3 expression in CTb-labeled projection neurons, two overlapping image stacks (16–31 optical sections of 1 or 2 μm thickness) were acquired with a ×10 (UPlanSApo, Olympus, N.A. 0.4) lens to include the entire dorsal horn. Retrogradely-labeled projection neurons were counted within laminae I–IV of the spinal dorsal horn. 

For immunofluorescent studies, the channel corresponding to p-S10H3 was viewed first (without a signal the sample was useless for analysis) followed by the other channels. System settings (laser power, confocal aperture and gain) were identical for the experiments when ipsi- and contralateral sides were compared.

Co-localization between p-S10H3 and neurochemical or reporter markers (such as tdTomato) was checked in a 100 µm-thick band from the surface of the SDH, corresponding to laminae I–IIo, where p-S10H3 nuclei were most abundant, by using the spots module of the Imaris image analysis software (BitPlane, version 8.1.0; Zurich, Switzerland) in surpass mode. This approach allowed one to identify all cells within the entire image volume in 3D defined by up to 45 optical sections at 0.5 µm z-separation. The DAPI-stained nuclei were used to unequivocally separate individual cells and to avoid counting the same neuron twice. The region of interest, laminae I and IIo, where p-S10H3 was clustered (a roughly 280 µm × 100 µm band) was isolated by cropping from the original image for further analysis. The channel corresponding to the p-S10H3 was checked and analyzed first, followed by the other channels one by one. Particles in each channel (DAPI-stained nuclei, immunoreactive dots, stained cytoplasm, etc.) were detected and fitted with spots by the automatic detection algorithm of the Imaris software. Filters used by the detection module (labeling quality, intensity center from the DAPI signal, intensity mean for the given channel) were manually adjusted when the default settings did not detect the whole range of particles of interest, since the chosen ROIs always contains manageable numbers of cells per sections. With the set parameters, the software module automatically detected all particles within a given channel. Colocalization/proximity between particles in different channels was calculated using a threshold distance of three. The total number of spots/ROI for each channel and number of colocalized spots between two channels were averaged (mean ± SEM) in Microsoft Excel. Finally, colocalization was expressed as a percentage.

### 4.8. Antibody Characterization

In the present study, two p-S10H3 antibodies (both from Abcam, Cambridge, UK; Table 4) were used to localize p-S10H3 nuclei in the SDH of mice and rats. Both antibodies are validated as ChIP-grade quality and have been referenced in more than 120 publications so far according to company data sheets. We thoroughly compared sections which were reacted with the two anti-p-S10H3 antibodies and found that they resulted in a close-to-identical burn injury-induced p-S10H3 expression pattern in the mouse dorsal horn (Appendix A). Specificity of the p-S10H3 antibodies was confirmed by their colocalization patterns in the ipsilateral side of burn injury and the lack of immunolabeling on the contralateral side of spinal cord (Figure 1b and Appendix A). The p-S10H3 antibody produced in rabbits was used in combination with most other primary antibodies (listed in Table 4) throughout the study except for those that were also raised in rabbits: anti-Pax-2, anti-Lmx1b and anti-somatostatin antibodies (see Table 4). In these cases, the p-S10H3 antibody produced in mice was used. A summary of all antibodies used in this study is shown in Table 4.

Specificity of the goat anti-CTb antibody was previously demonstrated by the lack of any staining in animals not injected with CTb [14,30]. The calcitonin gene-related peptide 1 (CGRP) antibody detects both α and β forms of the peptide [14,30]. NK1R-specific antibody was generated in guinea pigs using a peptide corresponding to the C-terminal end sequence of the rat NK1R [76]. Biotinylated IB4 was used to identify the IB4-binding subpopulation of non-peptidergic afferents [77]. The monoclonal antibody against calretinin reacts specifically with calretinin in tissue originating from human and rat as determined by its distribution in the brain, as well as by immunoblots [78]. Antibody against parvalbumin labels a subpopulation of neurons in the normal brain with high efficiency but does not stain the brain of parvalbumin knock out mice, according to the supplier. The polyclonal anti-prodynorphin antibody is a synthetic peptide (SQENPNTYSEDLDV) corresponding to amino acids 245–248 of rat Prodynorphin (Table 4). The polyclonal IBA1 antibody was raised against the synthetic peptide corresponding to AA 134 to 147 from rat IBA1. The monoclonal NeuN antibody was raised against cell nuclei purified from a mouse brain. The antibody against NPY was raised against a synthetic peptide (SDLLMRESTENAPRTR) corresponding to amino acids 76–91 of rat Neuropeptide Y. The antibody against nNOS was produced against a synthetic peptide with the sequence ESKKDTDEVFSS, representing the C-terminus of the human protein (residues 1423–1434) according to NP_000611. The polyclonal somatostatin antibody was raised against a synthetic peptide encompassing a sequence within the C-terminus region of the mouse somatostatin. The monoclonal substance P antibody was produced in rats with epitope mapping near the C-terminus of SP. The antibody against Pax-2 was a GST-Pax-2 fusion protein that was derived from the C-terminal domain (aa188-385) of the murine Pax-2 protein. The anti-Lmx1b antibody was a synthetic peptide within Human LMX1b aa 324–380 (C terminal). The chicken polyclonal antibody to GFP was raised against the recombinant full-length protein corresponding to GFP and it has been referenced more than in 1700 publications. The rat anti-RFP antibody was derived from the hybridoma supernatant. Omission control was routinely performed for all the antibodies.

### 4.9. Statistics

Statistical analyses were performed using Past4 software [79]. The Mann–Whitney U-test was applied to determine whether the proportion of p-S10H3-IR neurons differed significantly between the upper (L4–L6) and the lower (L1–L3) segments of the lumbar spinal cord. After normality tests, paired Student’s t-tests were used for comparing the average number of CTb-labeled neurons showing immunoreactivity to various markers (either CTb-IR alone or in combination with p-S10H3- or NK1R-IR) between the ipsi- and contralateral side of the burn injury. The Mann–Whitney U-test was also utilized to determine whether the proportion of various populations of p-S10H3-expressing neurons differ from each other significantly. Data is expressed as mean ± standard error of mean unless otherwise stated.

## 5. Conclusions

For the first time, we identified a subset of excitatory dynorphinergic interneurons that are the most affected by burn injury-induced tissue damage and are consequently activated by upregulation of the phosphorylated histone H3-mediated pathway. Based on our results, we propose that this distinct population of excitatory dynorphinergic neurons showing p-S10H3 immunoreactivity after noxious heat might have a crucial role during the development of burn injury-associated central sensitization and heat hyperalgesia.

## Figures and Tables

**Figure 1 ijms-22-02297-f001:**
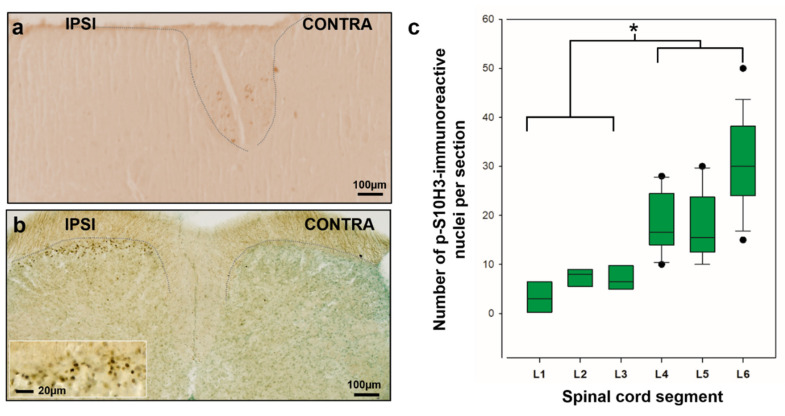
Burn injury-induced S10H3 phosphorylation is mainly located in the superficial dorsal horn (SDH) and displays unequal distribution within the spinal lumbar segments (L1–L6) in mice. (**a**) Innoxious stimulation (37 °C for 2 min) did not induce the phosphorylation of S10H3 in the SDH. (**b**) Burn injury (60 °C for 2 min) induced a robust phosphorylation of S10H3 in the ipsilateral SDH only. Both sections (**a**,**b**) were taken from L5 spinal segments. Dotted lines indicate the border between gray and white matter. (**c**) Burn injury-induced p-S10H3 expression shows unequal rostrocaudal distribution, with a peak expression in L6. Data presented as box plots are from 80-µm-thick sections taken from each segment (4–9 sections per segment from 3 animals). Dots indicate outlier values. * *p* < 0.05.

**Figure 2 ijms-22-02297-f002:**
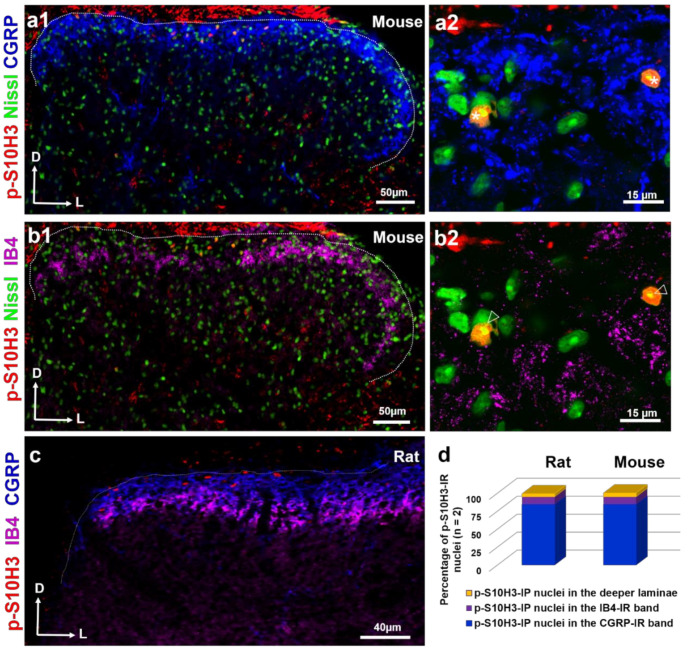
Neurons in the superficial dorsal horn exhibiting p-S10H3 immunoreactivity after burn injury show close appositions with CGRP-containing peptidergic afferents. (**a1**,**a2**) Immunostaining with antibodies against p-S10H3 (red), CGRP (blue) and Nissl (green). (**a2**) Asterisks indicate p-S10H3-positive neurons presenting close appositions with CGRP-containing peptidergic terminals. (**b1**,**b2**) Immunostaining with antibodies against p-S10H3 (red), isolectin B4 (IB4) (purple) and Nissl (green). (**b2**) None of the three p-S10H3-immunopositive neurons (empty arrowheads) have IB4 terminals in their vicinity. (**c**) Immunostaining with antibodies against p-S10H3 (red), CGRP (blue) and IB4 (purple). Dotted lines indicate the border between gray and white matter. D, dorsal; L, lateral. (**d**) Distribution of p-S10H3-immunoreactive nuclei within the CGRP- and IB4-positive bands and also in the deeper laminae of the SDH of rats and mice (n = 2 animals per species). The CGRP band contains the vast majority of p-S10H3 nuclei in both species.

**Figure 3 ijms-22-02297-f003:**
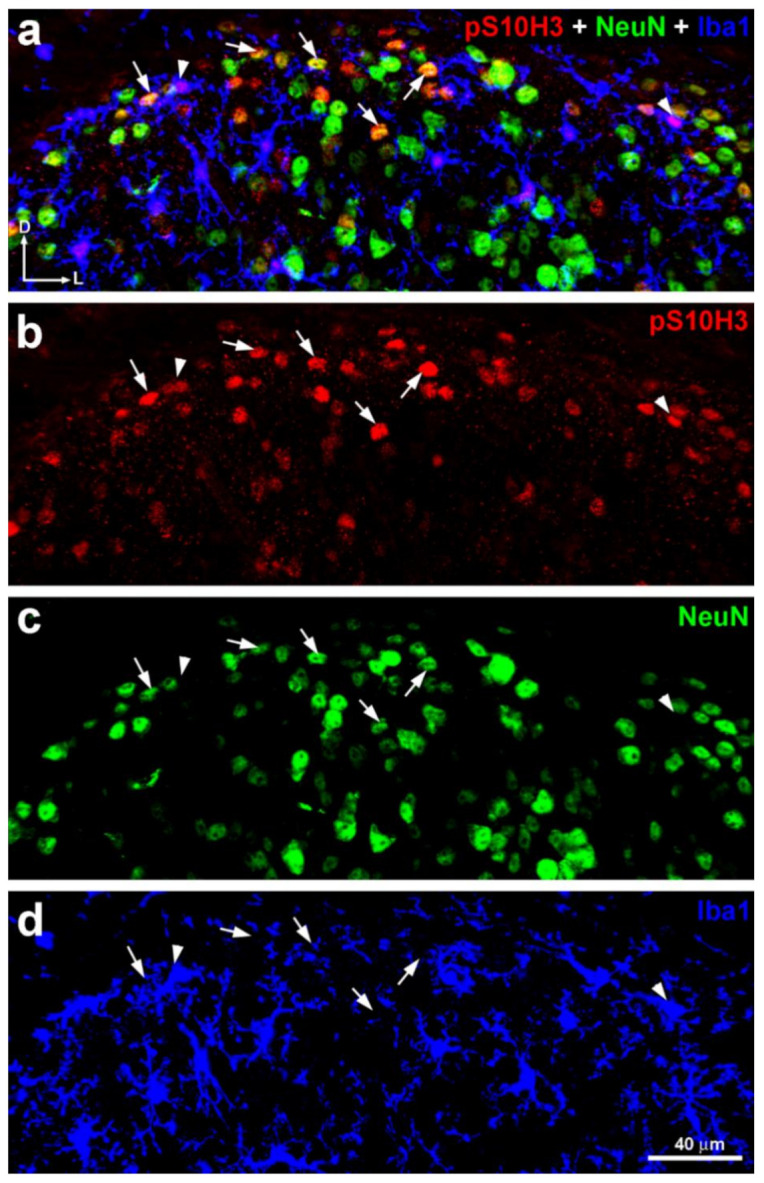
p-S10H3 was upregulated mainly but not exclusively in SDH neurons in mice after burn injury. (**a**–**d**) Immunostaining with antibodies against p-S10H3 (red; **b**), Fox-3 (NeuN) (green; **c**), and Iba-1 (blue; **d**) in a transverse section of wild-type mouse. The merged image (**a**) clearly shows that almost all p-S10H3 positive nuclei are located in NeuN-positive somata (arrows) indicating that majority of them belong to neuronal population. Arrowheads label p-S10H3-expressing microglial cells. D, dorsal; L, lateral.

**Figure 4 ijms-22-02297-f004:**
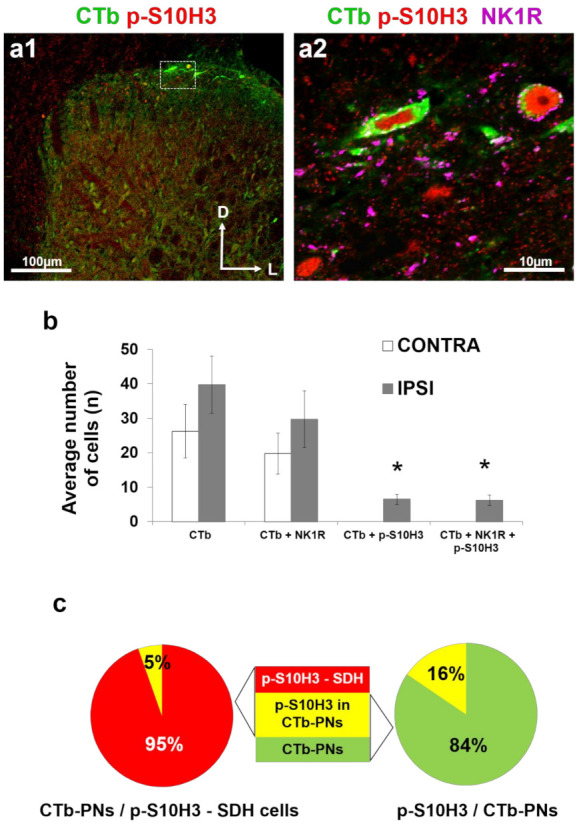
cholera toxin b-subunit (CTb)-labeled projection neurons rarely show nuclear burn injury-induced p-S10H3. (**a1**,**a2**) Immunostaining with antibodies against p-S10H3 (red), CTb (green) and neurokinin-1 receptor (NK1) receptor (NK1R; magenta) in an L5 transverse spinal cord section of a rat after burn injury. (**a2**) Higher magnification image represents the region with two CTb-labeled projection neurons (PNs) outlined by dashed line in (**a1**). These two CTb/p-S10H3 co-labeled neurons express NK1R. (**b**) Average number of CTb-labeled projection neurons (PNs) containing p-S10H3 positive nuclei and NK1R immunoreactivity (n = 17, 80-μm-thick sections per rat from 4 rats). IPSI refers to the side of the burn injury. Asterisks indicate a statistically significant difference between the ipsi- and contralateral sides of SDH (* *p* < 0.05). Note that the numbers of CTb-PNs with p-S10H3 or CTb/p-S10H3/NK1R ipsilaterally are negligible. Note also that the CTb-PNs could be detected on both sides. (**c**) Percentage of CTb-PNs within the total p-S10H3 positive SDH cell population (left) and percentage of PNs with p-S10H3 positive nuclei within the total CTb labeled PN population in the SDH (right).

**Figure 5 ijms-22-02297-f005:**
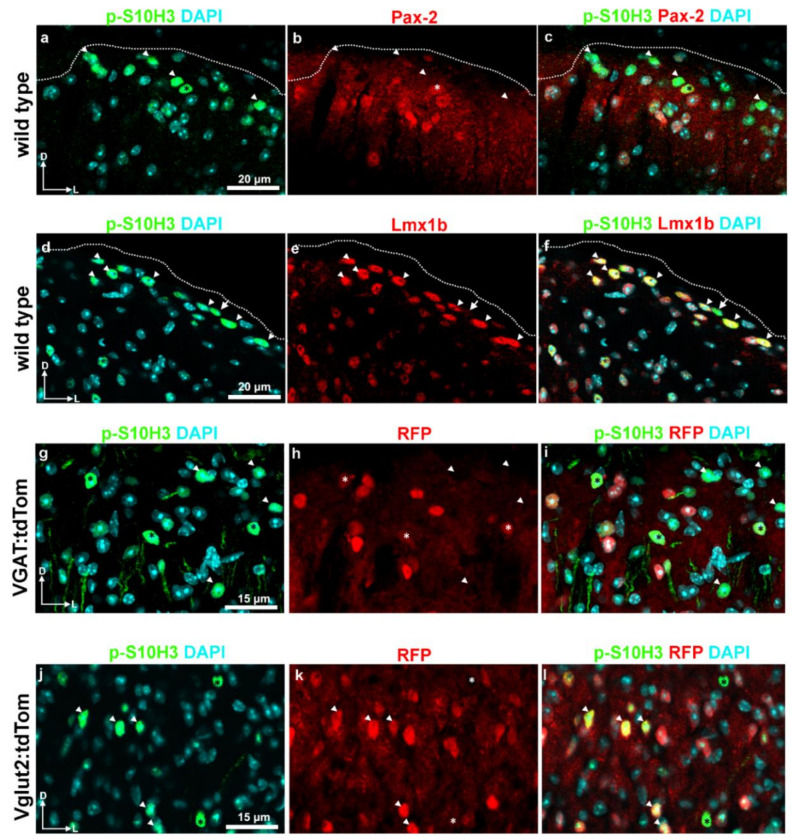
p-S10H3 is upregulated predominantly in excitatory interneurons both in wild-type and transgenic mice after burn injury. (**a**–**c**) Immunostaining with antibodies against p-S10H3 (green) and Pax-2 (red) in a transverse section of a wild-type mouse. In this field of view, there is only one Pax-2-positive neuron that exhibits p-S10H3-IR in its nucleus (asterisk). Arrowheads mark p-S10H3-IR neurons which lack Pax-2. (**d**–**f**) Immunostaining with antibodies against p-S10H3 (green) and Lmx1b (red) in a transverse section of a wild-type mouse. In this field of view, there are numerous p-S10H3-positive neurons co-labeled with Lmx1b (arrowheads), while there is only one p-S10H3-IR cell that lacks Lmx1b (arrow). The dotted line indicates the border between gray and white matter (**a**–**f**). (**g**–**i**) Immunostaining with antibodies against p-S10H3 (green) and red fluorescent protein (RFP) (red; against to tdTomato) in a transverse section of a vesicular gamma-amino butyric-acid (GABA) transporter (VGAT):tdTomato transgenic animal. There are three p-S10H3-positive neurons that exhibit weak RFP labeling in their cytoplasm (asterisks). Several p-S10H3-IR neurons without RFP-labeling can also be visible (arrowheads). (**j**–**l**) Immunostaining with antibodies against p-S10H3 (green) and RFP (red; against to tdTomato) in a transverse section of a Vglut2:tdTomato transgenic animal. Nearly half of the p-S10H3-positive nuclei are located in tdTomato-positive somata (arrowheads), indicating their glutamatergic nature. There are additionally two p-S10H3-expressing neurons that lack RFP signals (marked with asterisks). D, dorsal; L, lateral.

**Figure 6 ijms-22-02297-f006:**
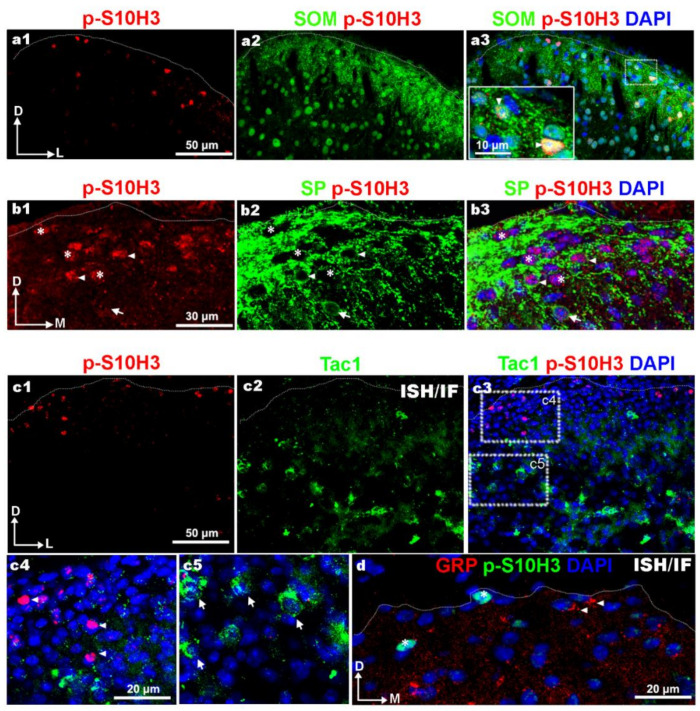
Burn injury-induced S10H3 phosphorylation in excitatory subpopulations of SDH neurons. (**a1**–**a3**) A representative image showing immunostaining for p-S10H3 (red; **a1**,**a3**) and somatostatin (SOM; green; **a2**,**a3**) in a transverse section of wild-type mouse. (**a3**) shows a merged image with 4′,6-diamidino-2-phenylindole (DAPI) (blue). In the inset there are two p-S10H3-positive neurons that exhibit strong SOM labeling in their cytoplasm (arrowheads). (**b1**–**b3**) A representative image showing immunostaining for p-S10H3 (red; **b1**, **b3**) and substance P (SP; green; **b2**,**b3**) in a transverse section of wild-type mouse. (**b3**) shows a merged image with DAPI (blue). There are some SP-containing neuronal soma with (arrowheads) or without p-S10H3 (arrow) in their nuclei. Asterisks show p-S10H3-IR neurons that probably do not produce SP. (**c1**–**c5**) A representative image showing immunostaining for p-S10H3 (red; **c1**,**c3**) and in situ hybridization (ISH) signal for *Tac1* mRNA (green; **c2**,**c3**). (**c3**) shows a merged image with DAPI (blue). Insets represent higher magnification views of regions of interest designated on image (**c3**). Arrowheads label p-S10H3-IR neurons that probably do not produce substance P due to its low level of *Tac1* mRNA (**c4**). Arrows indicates *Tac1* mRNA containing cell bodies that do not show p-S10H3-IR (**c5**). (**d**) Immunostaining for p-S10H3 (green), cell nuclei specific DAPI (blue) and ISH signal for GRP mRNA (red). Interestingly, SDH neurons containing GRP mRNA (arrowheads) never express pS10H3-positive nuclei (asterisks) following burn injury. The dotted line indicates the border between gray and white matter (**a**–**d**). D, dorsal; L, lateral; M, medial.

**Figure 7 ijms-22-02297-f007:**
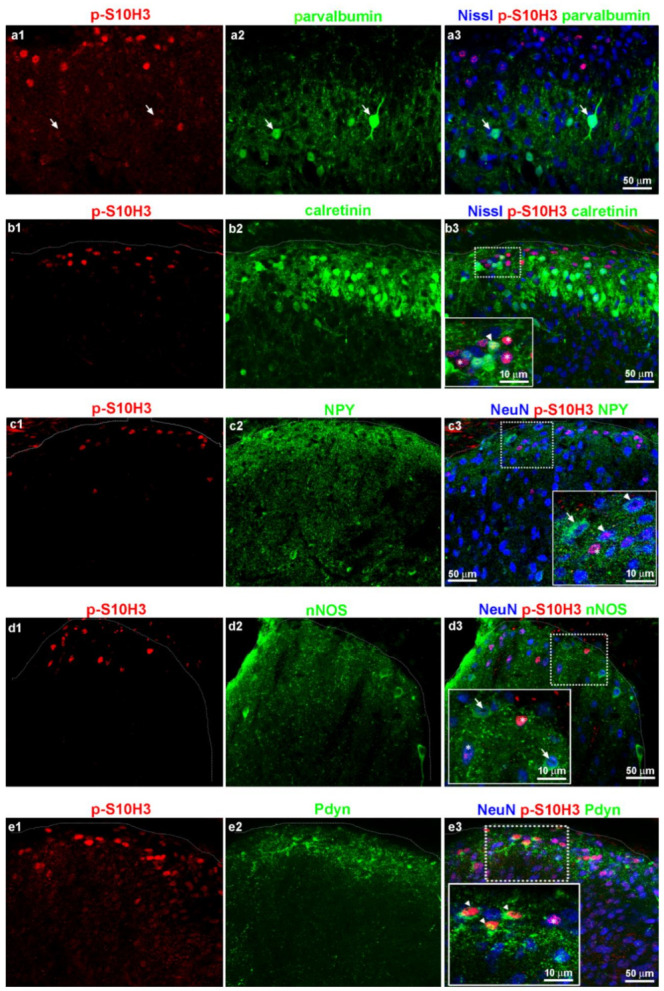
Burn injury-induced S10H3 phosphorylation in inhibitory subpopulations of SDH neurons. (**a1**–**a3**) A representative image showing immunostaining for p-S10H3 (red; **a1**,**a3**), Nissl dye (blue; **a3**) and parvalbumin (green; **a2**–**a3**). Nearly all p-S10H3-positive nuclei fell outside the band rich in parvalbumin-immunoreactive profiles and none was present in parvalbumin positive somata of deeper laminae (arrows). (**b1**–**b3**) Immunostaining for p-S10H3 (red; **b1**,**b3**), Nissl dye (blue; **b3**) and calretinin (green; **b2**,**b3**). Calretinin positive neurons occasionally contained p-S10H3-positive nuclei (arrowhead in the inset). Asterisks indicate p-S10H3-IR neurons without calretinin immunolabeling. (**c1**–**c3**) Immunostaining for p-S10H3 (red; **c1**,**c3**), neuronal marker, NeuN (blue; **c3**) and NPY (green; **c2**,**c3**). Hardly any of the rare NPY-IR neuronal somata contained a p-S10H3-positive nucleus as indicated with arrowheads. Asterisk marks p-S10H3-IR neurons without NPY immunolabeling. Arrow marks an NPY+ interneuron that lacks p-S10H3. (**d1**–**d3**) Immunostaining with antibodies against p-S10H3 (red; **d1**,**d3**), NeuN (blue; **d3**) and nNOS (green; **d2**,**d3**). Asterisk indicates a p-S10H3-IR neuron without nNOS immunolabeling. Arrows show nNOS-IR neurons in which p-S10H3 was not upregulated upon burn injury. (**e1**–**e3**) Immunostaining with antibodies against p-S10H3 (red; **e1**,**e3**), NeuN (blue; **e3**) and prodynorphin (Pdyn) (green; **e2**,**e3**). Several p-S10H3-immunostained nuclei were localized within dynorphin-positive somata (arrowheads in the inset). In the higher magnification image, an asterisk labels a non-dynorphinergic neuron that exhibits p-S10H3. Dotted lines indicate the border between gray and white matter (**b3,c3,d3,e3**).

**Figure 8 ijms-22-02297-f008:**
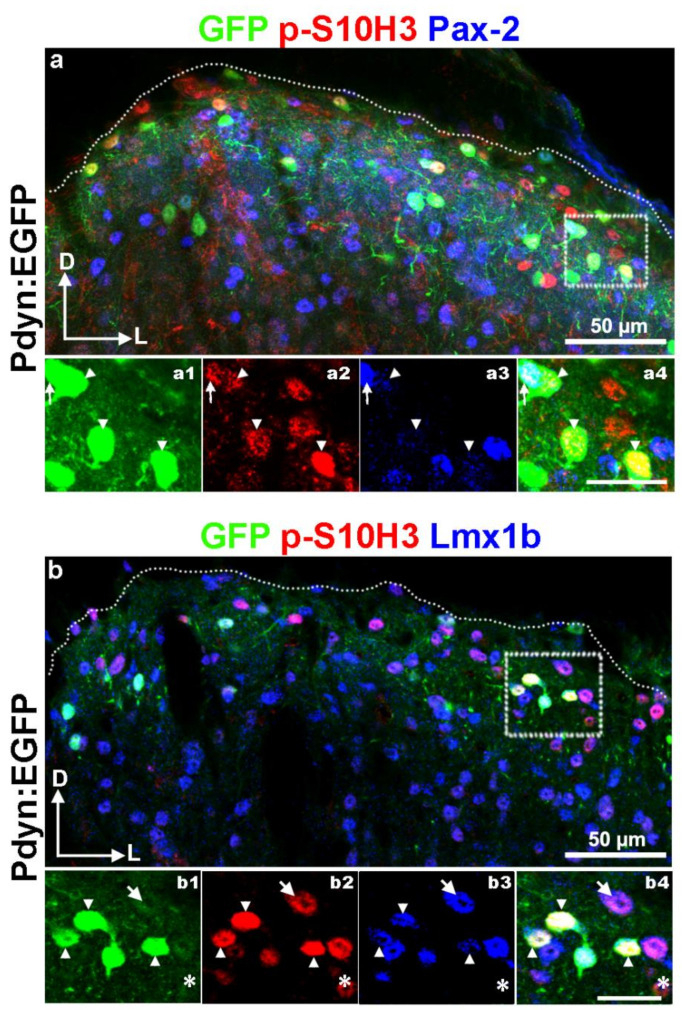
Burn injury-induced p-S10H3 nuclei are dominantly expressed by the excitatory subpopulation of dynorphinergic SDH neurons. (**a**) A representative image showing immunostaining for p-S10H3 (red), GFP (green) and Pax-2 (blue) in a transverse spinal cord section of Pdyn:enhanced green fluorescent protein (EGFP) mouse. In the inset (**a1**–**a4**) there are several anti-GFP antibody-labeled dynorphinergic neurons. Of these, there is only one p-S10H3-expressing Pdyn+ neuron that exhibits Pax-2-IR (labeled with an arrow), while the others lack Pax-2 (arrowheads; **a1**–**a4**). (**b**) A representative image showing immunostaining for p-S10H3 (red), GFP (green) and Lmx1b (blue) in a transverse spinal cord section of Pdyn:EGFP mouse. In the insets (**b1**–**b4**), numerous p-S10H3-immunoreactive nuclei are visible in laminae I–IIo of the SDH. Of these, the majority express Lmx1b and thus presumably they are excitatory. While there are several p-S10H3+/Lmx1b+ neurons that proved to be dynorphinergic based on their strong cytoplasmic GFP-staining (arrowheads in the insets), there is only one that is non-dynorphinergic (indicated by an arrow). The asterisk indicates such a p-S10H3-IR neuron that is immunonegative both for Lmx1b and Pdyn. The dotted line represents the border between gray and white matter (**a**,**b**). D, dorsal; L, lateral. Scale bars in the insets are 20 µm.

**Table 1 ijms-22-02297-t001:** Evaluation of the total number of p-S10H3-positive neurons in each lumbar segment (L6–L1) of mouse spinal cord following burn injury (n = 3 mice).

	L6	L5	L4	L3	L2	L1
Mean number of p-S10H3-positive neurons ^#^	30.7 ± 3.1	17.7± 2.7	18.5 ± 2.3	7.5 ± 1.5	7.5 ± 0.8	3.25 ± 2.7
Thickness of a given segment in mm *	1.45	1.18	1.31	1.45	1.58	1.31
Number of 80 μm-thick sections per segment ^£^	18.1	14.7	16.3	18.1	19.2	16.3
Total number of p-S10H3+ neurons per segment ^€^	557 ± 57	260.9 ± 40	301.5 ± 38	135.7 ± 28	144 ± 17	52.9 ± 44

^#^ calculated from 2–4 sections per 3 mice. * based on a study from Harrison et al. [27]. ^£^ thickness of a given segment was divided by the thickness of sections (80 μm). ^€^ total number of p-S10H3-positive neurons per segment was calculated as follows: mean number of p-S10H3-positive neurons (±SEM) was multiplied by the number of 80 μm thick sections per segment.

**Table 2 ijms-22-02297-t002:** Percentage of p-S10H3+ neurons after burn injury that were Pax-2 or Lmx1b-immunoreactive (IR) in wild type mice or that were VGAT- or vesicular glutamate transporter 2 (Vglut2)-IR in VGAT- or Vglut2:tdTomato transgenic mice.

	p-S10H3 ^#^	IR ^##^	Double ^###^	Double/p-S10H3	Double/IR
Pax-2	150	198	36	24.00%	18.10%
(n = 6)					
Lmx1b	297	711	219	73.70%	30.80%
(n = 9)					
VGAT	420	538	67	15.90%	12.40%
(n = 16)					
Vglut2	287	831	128	44.50%	15.40%
(n = 16)					

^#^ total number of cells with p-S10H3-nuclei 5 min after burn injury counted in lamina I and IIo. ^##^ total number of cells immunoreactive (IR) for the given cell type specific marker counted in lamina I and IIo. ^###^ total number of cells immunoreactive for the given marker showing p-S10H3 in their nuclei counted in lamina I and IIo. Number of sections (from 3 wild-type CD1 mice; two VGAT:tdTomato or 3 Vglut2:tdTomato transgenic mice) is shown in brackets. Values in the last column indicate % of p-S10H3-positive neurons that colocalized with the given marker.

**Table 3 ijms-22-02297-t003:** Proportion of p-S10H3 nuclei after burn injury in different subsets of neurons defined by their neurochemical phenotypes in laminae I and IIo. nNOS: neuronal nitrogen monoxide synthase.

	p-S10H3 ^#^	IR ^##^	Double ^###^	Double/p-S10H3	Double/IR
somatostatin	78	146	47	60.20%	32.10%
(n = 3)					
Dynorphin *	277	151	95	34.20%	62.90%
(n = 7)					
Dynorphin ^$^	202	186	78	38.60%	41.90%
(n = 10)					
calretinin	385	253	33	8.30%	12.60%
(n = 11)					
nNOS	248	72	8	3.20%	11.10%
(n = 7)					
NPY	154	6	1	0.60%	16.60%
(n = 5)					
parvalbumin	93	36	2	2.10%	5.50%
(n = 3)					

^#^ total number of neurons with p-S10H3-nuclei 5 min after burn injury. ^##^ total number of neurons immunoreactive (IR) for the given neurochemical marker. ^###^ total number of neurons immunoreactive for the given neurochemical marker showing p-S10H3 in their nuclei. * values from wild-type CD1 mice (n = 3). ^$^ values from Pdyn:EGFP transgenic mice (n = 2) in which dynorphinergic neurons express EGFP cre-dependently. Number of sections (from 3 CD1 mice; otherwise indicated) is shown in brackets. Values in the last column indicate % of p-S10H3-positive neurons that colocalized with the given neurochemical marker.

**Table 4 ijms-22-02297-t004:** Details of primary antibodies used for immunofluorescent staining.

Name	Species	Dilution	Supplier	Cat. No.
p-S10H3	rabbit	1:600	Abcam (Cambridge, UK)	ab5176
p-S10H3	mouse	1:1000	Abcam (Cambridge, UK)	ab14955
GFP	chicken	1:2000	Abcam (Cambridge, UK)	ab13970
RFP	rat	1:1000	Chromotec (Planegg-Martinsried, Germany)	5f8-100
Choleratoxin B	goat	1:2000	List Biol Labs (Campbell, CA, USA)	703
NK1R	guinea pig	1:25000	Merck, KGaA (Darmstadt, Germany)	AB15810
IBA-1	guinea pig	1:500	SYSY (Göttingen, Germany)	234 004
NeuroTrace 500/525 Nissl		1:1000	Thermo Fisher Sci. Inc. (Waltham, MA, USA)	N21480
CGRP	guinea pig	1:5000	Peninsula (San Carlos, CA, USA)	T-5027
Isolectin IB4 *		1:5000	Thermo Fisher Sci. Inc. (Waltham, MA, USA)	I21414
NeuN	mouse	1:1000	Millipore (Burlington, MA, USA)	MAB377
calretinin	mouse	1:6000	Swant (Marly, Switzerland)	6B3
parvalbumin	mouse	1:2000	Swant (Marly, Switzerland)	PV235
neuropeptid Y	guinea pig	1:600	Novus Biol (Centennial, CO, USA)	NB100-1624
nNOS	goat	1:1000	Novus Biol (Centennial, CO, USA)	NB100-858
somatostatin	rabbit	1:200	GeneTex (Irvine, CA, USA)	GTX133119
substance P	rat	1:100	Santa Cruz (Dallas, TX, USA)	sc-21715
Pax-2	rabbit	1:400	Thermo Fisher Sci. Inc. (Waltham, MA, USA)	71-6000
Lmx1b	rabbit	1:400	Abcam (Cambridge, UK)	ab66941
prodynorphin	guinea pig	1:600	GeneTex (Irvine, CA, USA)	GTX10280

* IB4, biotin-conjugated and visualized by fluorescent-streptavidin (1:1000, Thermo Fisher Sci. Inc. (Waltham, MA, USA)).

## Data Availability

The data presented in this study are available upon request.

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
