# Peer review of "Spinal Excitatory Dynorphinergic Interneurons Contribute to Burn Injury-Induced Nociception Mediated by Phosphorylated Histone 3 at Serine 10 in Rodents"

_ijms, 2021, doi:10.3390/ijms22052297_

Round 1

Reviewer 1 Report

  1. Number and exact date of issue of ethical agreement must be added into the study
  2. The exact age of animals, number of males and females as well as total number of animals used in the study must be added into the text
  3. The number of animals in each experimental group should be clearly identified in the text, for example in tabular form
  4. Information about city and country (in the case of the USA also the name of state) of producers of antibodies must be added in table 4
  5. Table 4 must be relocated to “Materials and methods”, to subchapter 4.5
  6. Whether the Authors performed a preabsorption test to verify specificity of antibodies. If not, why the Authors resigned from this test. If yes, the exact description with used antigens must be added. The reviewer knows that specificity tests has been done in previous works, but such test should be done in tissues used in the experiment.
  7. Why the number of animals in particular groups was differ?
  8. In opinion of the reviewer subchapters of materials and methods 4.5 and 4.8 should be connected. Now this part of the manuscript is not clear
  9. The adding of exact description of scale bars on microphotographs would increase readability of figures. Microphotographs in figures have various magnification and now reader is forced to find scale bars in long legends.
  1. Legends in figures are too long. Maybe they could be shortened?
  2. Some microphotographs are rather of poor quality (Fig2 a2, Fig 2b2, Fig 3c, Fig 7e – especially e1). Reviewer suggests to replace them with others
  3. Some observations have been made on only three animals. In opinion of the reviewer group of three animals is too small to draw right conclusions. Why the Authors decided to do experiment on such small number of animals. Why various parts of experiment have been conducted on various number of animals (for example some studies has been conducted on three animals, other on four animals)
  4. Conclusion that results obtained in present study can contribute to development of novel therapeutics for burn injury seems to be too far-reaching.

Author Response

Response to Reviewer #1: 

Thank you very much for your thorough and constructive refereeing. We are grateful for your comments. During the revision, we have addressed all your comments, concerns and questions and we are confident that, with your helpful refereeing, we have significantly improved the manuscript.

We hope that you will accept all of our answers, which you can read as follows.

  1. Number and exact date of issue of ethical agreement must be added into the study

We apologize for this shortcoming. The information is now given in the revised manuscript. (lines 867-869)

  1. The exact age of animals, number of males and females as well as total number of animals used in the study must be added into the text

We agree absolutely. The revised version of the manuscript now contains these details. (lines 866 & 888-890)

  1. The number of animals in each experimental group should be clearly identified in the text, for example in tabular form

We believe that the number of animals in each experiments has been indicated clearly both in the text and the corresponding tables (e.g. lines 612, 616 – number of animals in the experiment described in Figure 8.). We hope that including further details about the animal numbers and ages (see point 2. above) along with the already provided information leaves no doubt about the animal numbers.

  1. Information about city and country (in the case of the USA also the name of state) of producers of antibodies must be added in table 4

This information is now provided in the revised manuscript (see lines 932-936).

  1. Table 4 must be relocated to “Materials and methods”, to subchapter 4.5

We thank the referee for this constructive suggestion. We did the required modification (table 4 moved to subchapter  4.5.) and we feel that it improved our manuscript. (lines 932-936)

  1. Whether the Authors performed a pre-absorption test to verify specificity of antibodies. If not, why the Authors resigned from this test. If yes, the exact description with used antigens must be added. The reviewer knows that specificity tests has been done in previous works, but such test should be done in tissues used in the experiment.

We took care of using antibodies that all had been tested (see the corresponding subchapter in Methods and Materials) comprehensively in many different ways (by pre-absorption assays or involving knock-out animals) previously by others, using same conditions (tissue, preparation). We believe that re-testing and re-validating reliability of those antibodies would unnecessarily increase the number of animals and consumables. Regarding the p-S10H3-specific antibodies applied, both have been validated as ChIP-grade quality and referenced in more than 120 publications so far according to company data sheets. Nevertheless, we felt that as key parts of our study these antibodies should be further validated. Thus, we thoroughly compared sections which had been reacted with the two anti-p-S10H3 antibodies and found that they resulted in a close-to-identical burn injury-induced p-S10H3 expression pattern in the mouse dorsal horn (Suppl. Fig.S3). Specificity of the p-S10H3 antibodies was confirmed by their colocalization patterns in the ipsilateral side of burn injury and the lack of immunolabeling on the contralateral side of spinal cord (Suppl. Fig.S3). In addition, according to the manufacturer neither recognizes the non-modified histone as assessed by Western blot (no blocking had been seen with the non-phospho peptides). Unfortunately, we could not confirm this in own experiments, since absorption control (i.e. human histone H3 phosphoS10 peptide for ab14955), to the best of our knowledge and based on information received from the distributor, is not available for blocking p-S10H3.

  1. Why the number of animals in particular groups was differ?

The answer to this question is joined with the one given for point 12. raised by the referee. See below.

  1. In opinion of the reviewer subchapters of materials and methods 4.5 and 4.8 should be connected. Now this part of the manuscript is not clear

While the information in these two subchapters are closely related we believe that the general structure of our manuscript (i.e. details are discussed separately) is better followed of we keep these subchapters separate.

  1. The adding of exact description of scale bars on microphotographs would increase readability of figures. Microphotographs in figures have various magnification and now reader is forced to find scale bars in long legends.

We agree with the suggestion and during the revision we have modified the images accordingly.

  1. Legends in figures are too long. Maybe they could be shortened?

We absolutely agree with this comment and during the revision we tried to shorten figure legends as much as possible.

  1. Some microphotographs are rather of poor quality (Fig2 a2, Fig 2b2, Fig 3c, Fig 7e – especially e1). Reviewer suggests to replace them with others

Fig2 a2 and b2 are high magnification images. We replaced them with images that are now acceptable both in terms of signal-to-noise ratio, dynamic range and resolution.

Fig 3c shows an intense Iba1 immunoreactive signal in blue (pseudocolor) and the referee is right that the image could be improved in terms of dynamic range. Thus, we modified the image and increased the ROI to provide better orientation for the reader.

We have changed the whole panel e in Figure 7 with images where addition of more optical confocal sections in the Z-projected image improved the intensity of the p-S10H3 signal in cell nuclei.

  1. Some observations have been made on only three animals. In opinion of the reviewer group of three animals is too small to draw right conclusions. Why the Authors decided to do experiment on such small number of animals. Why various parts of experiment have been conducted on various number of animals (for example some studies has been conducted on three animals, other on four animals)

We made an effort to minimize the number (and also possible distress) of animals involved in the study to comply with 3R regulations. The number of animals for a given marker might differ as it was determined based on how abundant the particular marker is in the dorsal horn. We also believe that in case of inbred strains biological variability between animals is less, so selecting the minimal number of animals and increasing the number of sections or regions/sections provides a better overall picture while avoids unnecessary animal use. We believe that the number of animals selected for each analysis was sufficient for the required subsequent statistical analysis (to obtain valid statistical data).

  1. Conclusion that results obtained in present study can contribute to development of novel therapeutics for burn injury seems to be too far-reaching.

We agree that this conclusion – reflecting our optimistic enthusiasm for our work – is a bit far-reaching. Thus, in the revised manuscript we removed this art of our conclusion (from lines 1051-1055).

In addition, the revised version of the manuscript has undergone extensive grammar correction.

Reviewer 2 Report

The paper is well written and designed, the experiments were performed in a rigorous manner.

The paper needed the english extensive revision.

Author Response

We thank Reviewer #2 for his positive assessment.

The revised version of the manuscript has undergone extensive grammar correction.

Round 2

Reviewer 1 Report

All suggestions have been taken into account. The manuscript may be published in the present form